# Next generation proton PDFs
# with deuteron and nuclear uncertainties

**Rosalyn L. Pearson**⋆, **Richard D. Ball** and **Emanuele R. Nocera**

The Higgs Centre for Theoretical Physics, University of Edinburgh,
JCMB, KB, Mayfield Rd, Edinburgh EH9 3JZ, Scotland

⋆ r.l.pearson@ed.ac.uk

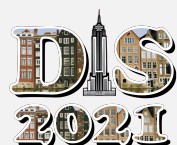 *Proceedings for the XXVIII International Workshop
on Deep-Inelastic Scattering and Related Subjects,
Stony Brook University, New York, USA, 12-16 April 2021*

## Abstract

As data become more precise, estimating theoretical uncertainties in global PDF determinations is likely to become increasingly necessary to obtain correspondingly precise PDFs. Here we present a soon-to-be-released next generation of global proton PDFs (NNPDF4.0) that include theoretical uncertainties due to the use of heavy nuclear and deuteron data in the fit. We estimate these uncertainties by comparing the values of the nuclear observables computed with the nuclear PDFs against those computed with proton PDFs. For heavy nuclear PDFs we use the nuclear nNNPDF2.0 set, while for deuteron PDFs we develop an iterative procedure to determine proton and deuteron PDFs simultaneously, each including the uncertainties in the other. Accounting for nuclear uncertainties resolves some of the tensions in the global fit of the proton PDFs, especially those between the nuclear data and the extended LHC data set used in NNPDF4.0.

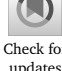
## 1  Introduction

Accounting for uncertainties due to nuclear effects is an important component in attempts to determine proton parton distribution functions (PDFs) to an accuracy of 1%. Nuclear effects have been considered many times before [1–5], and their impact on NNPDF3.1 fits using the theory covariance matrix formalism [6,7] was discussed in [8] and [9]. Here we show the impact of nuclear uncertainties in the upcoming release, NNPDF4.0: next generation PDFs which include deuteron and nuclear uncertainties by default.

The nuclear data to be included in NNPDF4.0 are shown in Table 1. There will be ∼4000 data points in total in the NNPDF4.0 fit, with roughly 10% being deuteron data and 20%

Table 1: The nuclear data to be included in NNPDF4.0. The process (Deep inelastic scattering (DIS) charged current (CC), neutral current (NC) and Drell-Yan (DY) is displayed for each dataset, alongside the number of data ($N_{dat}$) and the target.

| Nuclear data | | | |
|---|---|---|---|
| **Dataset** | **Process** | $N_{dat}$ | **Target** |
| DYE605 [10] | DY | 85 | $^{64}_{32}$Cu |
| NuTeV [11] | DIS CC | 76 | $^{56}_{26}$Fe |
| CHORUS [12] | DIS CC | 832 | $^{208}_{82}$Pb |
| SLAC [13] | DIS NC | 67 | $^2H$ |
| BCDMS [14] | DIS NC | 581 | $^2H$ |
| NMC [15] | DIS CC | 204 | $^2H$ and $p$ |
| DYE866/NuSea [16] | DY | 15 | $^2H$ and $p$ |
| DYE906/SeaQuest [17] | DY | 6 | $^2H$ and $p$ |

being heavy nuclear data. These are the same data [10–16] that were used in NNPDF3.1 with the addition of the recent SeaQuest data [17]. We can determine nuclear corrections by comparing the theory predictions for nuclear observables using proton PDFs with those using nuclear PDFs. The shift between the predictions quantifies the size of nuclear correction for that observable. The collective shifts can then be used to construct a theory covariance matrix [6]. We look separately at heavy nuclear corrections (for Cu, Fe and Pb) and deuteron corrections. This is because deuterons, being only a proton and a neutron, are qualitatively different from a heavy nuclear environment such as $^{56}$Fe, with 26 protons and 30 neutrons bound together. For the heavy nPDFs we used the NLO nNNPDF2.0 determination [18]. For the deuteron PDFs we developed a self-consistent iterative procedure [9] at NNLO within the NNPDF4.0 formalism.

## 2 Deuteron and heavy nuclear uncertainties

We include nuclear uncertainties using the covariance matrix methodology previously developed in NNPDF [6]. The contributions to the covariance matrix are determined following [8] and [9] based on the difference between calculating the nuclear observables with proton PDFs and calculating them with nuclear PDFs. In these papers we describe two approaches: including an uncertainty only ("deweighted" procedure); and also shifting the nuclear observable predictions ("shifted" procedure). We consider both these approaches here, with the deweighted one to be the default in NNPDF4.0.

The per-point uncertainties (square root of diagonal elements of covariance matrices) are shown in Fig. 1 as a % of the data. The deuteron uncertainties are for the most part a few percent, while the uncertainties for heavy nuclei are typically larger, ranging from 5% to 30% or more. The shape of the uncertainties is highly dependent on kinematics, with the increased uncertainty in the high $x$ nuclear shadowing region reflecting the larger nuclear effects there. The impact of including nuclear uncertainties is therefore minimal for the deuteron data but significant for heavy nuclear, so the latter will be deweighted more in the fit.

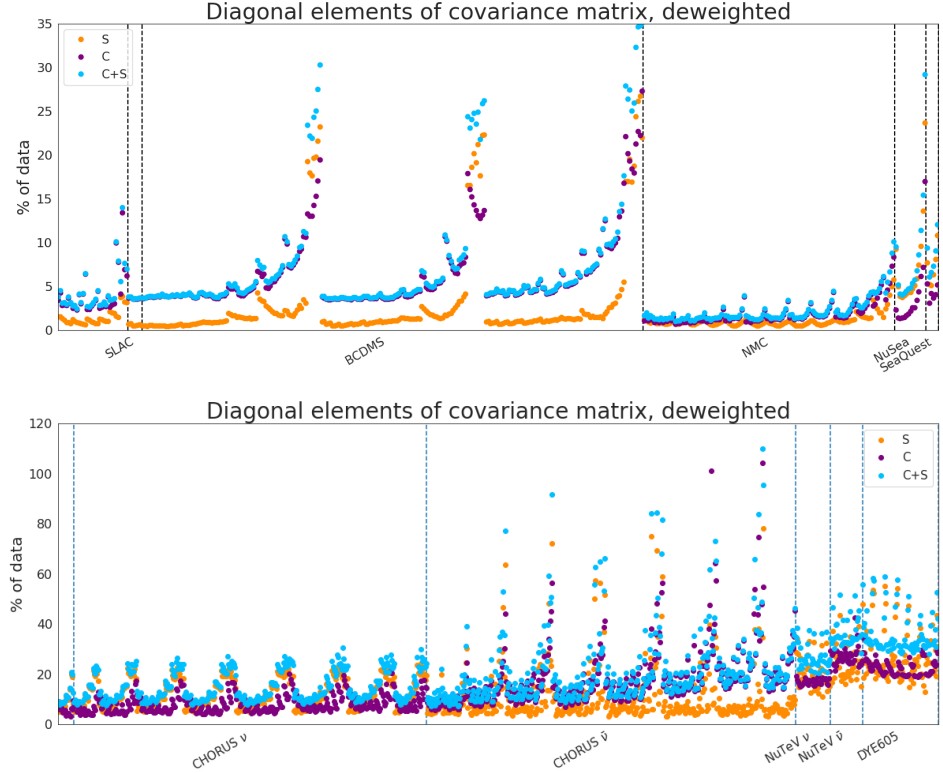

Figure 1: Square root of diagonal elements of covariance matrices for experimental $C$ (purple), nuclear $S$ (orange) and total $C + S$ (blue). Data are arranged in $Q^2$ bins with increasing $x$ in each bin. All values are displayed as a % of data. Top: deuteron, bottom: heavy nuclear. We only display results for the deweighted procedure; those for the shifted procedure are qualitatively similar.

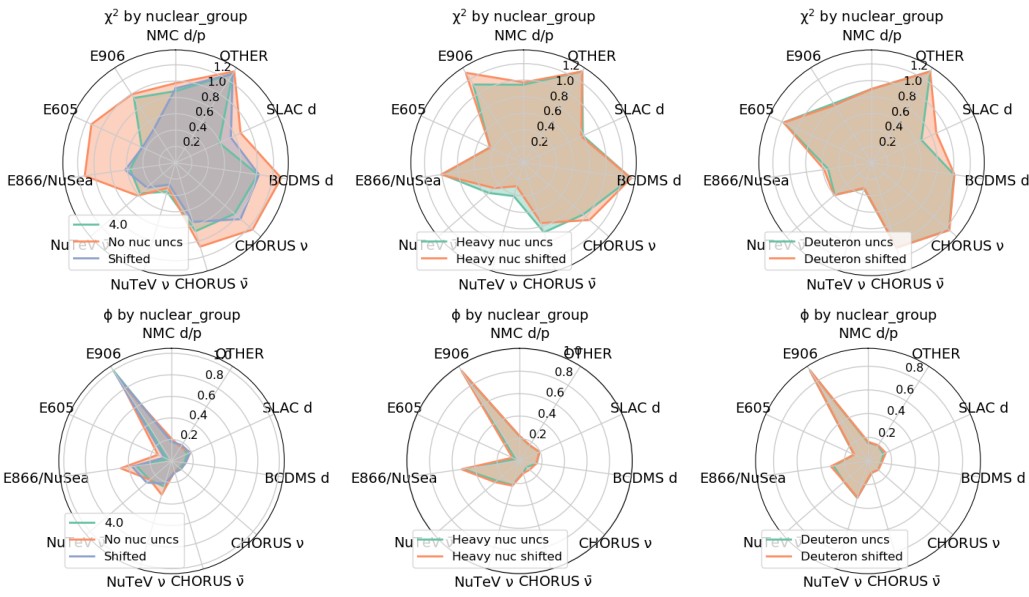

Figure 2: Partial $\chi^2$ (top row) and $\phi$ (bottom row) values broken down by nuclear dataset for the different configurations of uncertainties. All other datasets are collected under OTHER.

Table 2: Total $\chi^2$ and $\phi$ values for nuclear data sets for the various fits. 4.0 is the baseline fit from NNPDF4.0, no nuc unc is the same without nuclear uncertainties, deuteron/heavy nuc unc are with deuteron/heavy nuclear uncertainties only, deuteron/heavy nuc shifted are the same but where the nuclear observable central values are also shifted.

|  | No nuc unc | Deuteron unc | Heavy nuc unc | **4.0** | Deuteron shifted | Heavy nuc shifted | Shifted |
|---|---|---|---|---|---|---|---|
| $\chi^2$ | 1.269 | 1.257 | 1.193 | **1.162** | 1.244 | 1.196 | 1.166 |
| $\phi$ | 0.160 | 0.158 | 0.160 | **0.164** | 0.158 | 0.170 | 0.169 |

## 3 PDFs with nuclear uncertainties

We performed fits with configurations of nuclear uncertainties and shifts. NNPDF4.0 includes deuteron and heavy nuclear uncertainties, both deweighted. Table 2 gives the total $\chi^2$ and $\phi$ values for these fits (which are described in the caption), where $\phi$ is defined

$$\phi \equiv \sqrt{\langle \chi^2[T]\rangle - \chi^2[\langle T\rangle]}, \tag{1}$$

where $T$ are the theoretical predictions and $\langle \cdot \rangle$ denotes the average over PDF replicas. In [19] it is shown that this gives the ratio of uncertainties after fitting to the uncertainties of the original data, averaged over data points. The partial values per nuclear dataset can be seen in Fig. 2. Overall, including nuclear uncertainties results in a reduction in the $\chi^2$ from 1.27 to 1.17, indicating a substantially better fit quality, accompanied by an increase in $\phi$ from 0.160 to 0.164 due to the increase in uncertainty. The impact at the nuclear dataset level is striking, with a significant improvement in both $\chi^2$ and $\phi$ for most of these datasets. The difference between the deweighted and shifted prescriptions, however, is minimal.

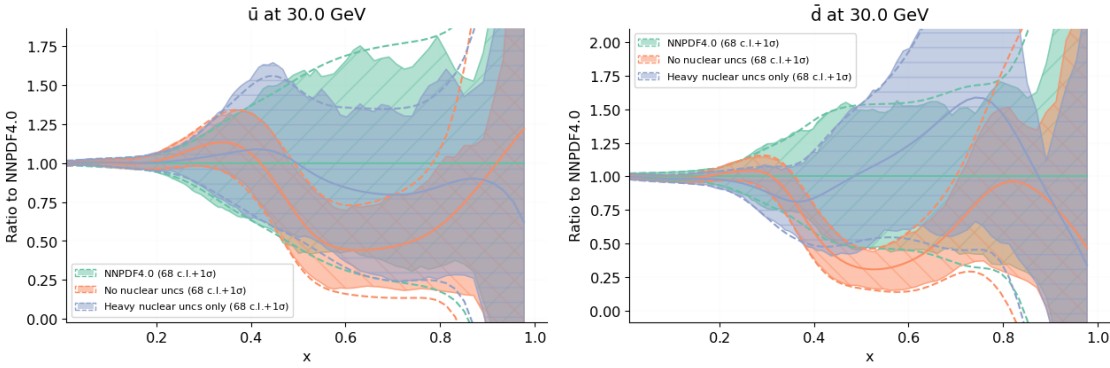

Figure 3: Impact of including nuclear uncertainties in NNPDF4.0. The default (green) is to include them for all nuclear data. Fits with no nuclear uncertainties (orange) and with only heavy nuclear uncertainties (blue) are also shown. We display the $\bar{u}$ and $\bar{d}$ distributions because the effect is greatest on these. The bands are 68% confidence intervals and the dotted lines are $1\sigma$.

At the PDF level, the impact is important at large $x$, as expected. Firstly, we want to know the impact of adding nuclear uncertainties. In Fig. 3 we compare NNPDF4.0 to a fit without nuclear uncertainties. We provide the $\bar{u}$ and $\bar{d}$ PDFs, which are representative of the rest of the flavours. Including uncertainties leads to significant change in the shape of the PDFs in the high $x$ shadowing region. Without nuclear uncertainties, the PDFs are pulled downwards towards the nuclear PDFs. In Fig. 3 we also show a fit with only heavy nuclear uncertainties. It is clear that the heavy nuclear uncertainties, being larger, produce the bulk of the impact.

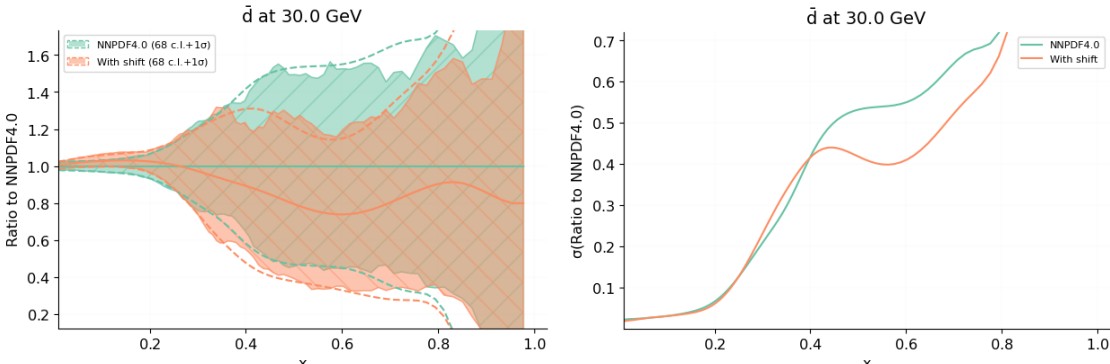

Figure 4: The impact of shifting (orange) versus deweighting (green) for the $\bar{d}$ distribution. The right panel shows the uncertainties for clarity. The effects on the $\bar{u}$ distribution are qualitatively similar. The bands are 68% confidence intervals and the dotted lines are $1\sigma$.

Secondly, we compare the deweighted and shifted approaches in Fig. 4. Using the shifted procedure clearly reduces the uncertainties relative to the deweighted one. However there is little impact at the level of $\chi^2$ and $\phi$ values, and the two outcomes are equivalent within uncertainties. From this we see that choosing one of these approaches over the other will not have a great impact. When making the choice of approach, we note that the shift is calculated relative to the value with proton PDFs, and so is itself dependent on the proton PDFs. This opens up the risk of double counting, and so the shift must be treated with caution. Adding uncertainties, however, always decreases the weight of data points, and so the deweighted prescription is the most conservative. Given also that it leads to a slightly lower total $\chi^2$ and $\phi$, we will opt to use the deweighted prescription in NNPDF4.0; including only an uncertainty for both deuteron and heavy nuclear data.

## 4  Conclusion

We used the theory covariance matrix formalism to include nuclear uncertainties in the upcoming release, NNPDF4.0, leading to a reduced global $\chi^2$. We see a modest shift in the central values of the PDFs and an increase in uncertainties in the high $x$ nuclear data region. This difference is driven mainly by the heavy nuclear data. We also investigated a procedure to shift the nuclear predictions, which was equivalent within uncertainties to including only an uncertainty, but with a slightly higher global $\chi^2$ and smaller PDF uncertainties. We opt to include uncertainties without a shift in NNPDF4.0.

**Funding information**    R.D.B. and E.R.N. are supported by the UK STFC grants ST/P000630/1 and ST/T000600/1.  E.R.N. was also supported by the European Commission through the Marie Skłodowska-Curie Action ParDHonSFFs.TMDs (grant number 752748).  R.L.P. is supported by the UK STFC grant ST/R504737/1.

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
