# Peer review of "Next generation proton PDFs with deuteron and nuclear uncertainties"

_SciPost Physics Proceedings, doi:SciPost Phys. Proc. 8, 026 (2022)_

## Round 1 · Referee Report · Anonymous (Referee 1) · 2021-7-17

Strengths

  1. Brief report on important theoretical uncertainty from the inclusion of heavy target and deuterium target data in PDF fits

Weaknesses

  1. At times too brief a report

Report

This is a conference report covering briefly material that has previously been published. As such it is somewhat short on detail, for example the \phi variable is hard to understand 'stand-alone' within this paper. However, it conveys the main results of taking into account theoretical uncertainties from deuterium and heavy target corrections on PDF fits, by two different methods. It should be published with some minor revisions.

Requested changes

  1. The notation for deuterium in Table 1 is wrong, I am sure the authors do not mean helium
  2. The paper reads as if NNPDF4.0 already exists. Perhaps it does, but only privately, within that collaboration. There has not yet been a full paper or submission to LHAPDF, only papers previewing it. Could the authors inidcate this in the language used. 3.Fig3, rhs showing the effect on dbar is hard to understand. If green is taking into account both deuterium and heavy target uncertainties, and red is not accounting for either, should not blue, which accounts for only heavy target uncertainties, be in between them in terms of its level of uncertainty and its central value? This does not seem to be the case, at least around x~0.6. Could the authors explain this?

  • validity: high
  • significance: good
  • originality: high
  • clarity: good
  • formatting: good
  • grammar: excellent

Author:  Rosalyn Pearson  on 2021-07-26  [id 1616]

(in reply to Report 1 on 2021-07-17)

We thank the referee for their comments on our proceedings, which we have addressed in the revised manuscript. Specifically we have addressed them as follows: 1. Thank you for spotting this incorrect notation, we have amended it in the table. 2. We have amended the language throughout to reflect that NNPDF4.0 has not yet been released. 3. The effect in Fig. 3 unfortunately does not have a simple interpretation. Note that when adding uncertainties for e.g. heavy nuclear data, we are deweighting the heavy nuclear data in the fit and therefore correspondingly increasing the weight of the other data, in particular the LHC data. Then when adding uncertainties for deuteron data, we deweight the deuteron data and increase the relative weight of everything else. Thus adding uncertainties relieves tension in the fit which can result in a complex selection of shifts in the PDFs, especially when one notes that they are also subject to constraints such as the sum rules. Hence we see in that for $\bar{u}$ the PDFs follow the pattern suggested by Referee 1, but the same is not true for $\bar{d}$.

---

## Round 1 · Referee Report · Anonymous (Referee 2) · 2021-7-20

Report

These proceedings provide a nice summary of recent work by the NNPDF collaboration on the inclusion of nuclear corrections in their fit. It is certainly suitable for publication. I have only a couple of minor suggestions, which the authors may wish to include:

1) Introduction section paragraph. Perhaps clarify explicitly that 'there are ~ 4000 data points in total in the NNPDF4.0 fit' - on first reading I was a little confused, as it seemed the 4000 referred to the number of nuclear data points.

2) Fig. 4 and discussion below Fig. 3. To me at least it is not too obvious by eye that the shifted procedure 'clearly' reduces the uncertainty relative to the deweighted. As always it can be a little hard to tell on these ratio plots. Given the breakdown between deuteron and heavy nuclear only is not discussed in this instance, it might be clearer to simply show the result with both nuclear and deuteron uncertainties, and both (shifted), as an absolute ratio but also with a separate plot just giving the corresponding relative errors.

  • validity: -
  • significance: -
  • originality: -
  • clarity: -
  • formatting: -
  • grammar: -

Author:  Rosalyn Pearson  on 2021-07-26  [id 1617]

(in reply to Report 2 on 2021-07-20)

We thank the referee for this report on our manuscript. We have addressed the comments as follows: 1. We have added the clarification suggested. 2. We agree that this was unclear from the plots shown and have adjusted the plots as recommended. Please find the uncertainties plot attached to this reply. It should hopefully now be clear that the uncertainties are reduced.

Attachment:

---

## Round 1 · Referee Report · Anonymous (Referee 3) · 2021-7-27

Strengths

  1. This is a summary of an update of an important element of the procedure for producing one of the small number of very widely used PDF sets. It describes what is undoubtedly an improvement to previous NNPDF analyses which largely did not take into account nuclear corrections.

  2. A large amount of information is conveyed in a small amount of space.

Weaknesses

  1. The summary is relatively brief, but this is forced upon it due to it being a conference proceedings with limited space.

  2. Due to brevity some aspects are not explained clearly and the appeal to references for detail is larger than ideal. However, again this is due to the constraints.

Report

The criteria are met very comfortably. As mentioned above this is a summary of a larger piece of work and some lack of detail is inevitable. I do not insist on any changes. However, I have some optional suggestions.

The conclusion repeats some detail and is perhaps longer than necessary for this format. A slight shortening could give room for a little more detail elsewhere. two specific suggestions would be:

Explain why $\bar u$ and $\bar d$ are appropriate PDF choices for Fig. 3.

I think it would be helpful to define, even approximately what $\phi$ represents, rather than appeal to a reference, as it is a central part of the results.

I also wonder if the authors have modified their views on the relative merits of the "shifting" versus "deweighted" approaches, given that the recent preprint on theoretical uncertainties by two of them, 2105.05114, modifies its conclusions somewhat upon revision and perhaps acknowledges more importance in the ability of the fit with theoretical uncertainties to improve the central predictions via learning? If so, the authors may wish to add/modify comments comparing "deweighting" to "shifting" and the relative merits.

Requested changes

Optional changes are given in the report above.

  • validity: high
  • significance: high
  • originality: high
  • clarity: good
  • formatting: good
  • grammar: excellent

Author:  Rosalyn Pearson  on 2021-08-02  [id 1629]

(in reply to Report 3 on 2021-07-27)

We thank Referee 3 for their comments on the proceedings, and have addressed their comments as follows:

  1. We have shortened the conclusion. $\bar{u}$ and $\bar{d}$ are appropriate choices of PDFs because they are indicative of the wider pattern of changes, and are influenced strongly by the nuclear data.
  2. We have included an equation with the definition of $\phi$
  3. We have added a short comment on choosing deweighted over shifted which is as follows: "Secondly, we compare the deweighted and shifted approaches in Fig.3. Using the shifted procedure clearly reduces the uncertainties relative to the deweighted one. However there is little impact at the level of $\chi^2$ and $\phi$ values, and the two outcomes are equivalent within uncertainties. From this we see that choosing one of these approaches over the other will not have a great impact. When making the choice of approach, we note that the shift is calculated relative to the value with proton PDFs, and so is itself dependent on the proton PDFs. This opens up the risk of double counting, and so the shift must be treated with caution. Adding uncertainties, however, always decreases the weight of data points, and so the deweighted prescription is the most conservative. Given also that it leads to a slightly lower total $\chi^2$ and $\phi$, we will opt to use the deweighted prescription in NNPDF4.0; including only an uncertainty for both deuteron and heavy nuclear data."

---

## Editorial Decision

published